# Experimental Study on Co-Firing of Coal and Brewery Wastewater Sludge

**Zixue Luo \*, Danxia Xu, Yanxiong Ma and Qiang Cheng**

State Key Laboratory of Coal Combustion, Huazhong University of Science and Technology, Wuhan 430074, China; m201771271@hust.edu.cn (D.X.); m202071264@hust.edu.cn (Y.M.); chengqiang@hust.edu.cn (Q.C.)

**\*** Correspondence: luozixue@hust.edu.cn; Tel.: +86-2787542417; Fax: +86-2787545526

**Abstract:** The environmental pollution and high energy consumption caused by the coal-dominated energy structure in China have been the focus of attention for a long time. The co-firing of biomass with coal can save coal resources and realize effective utilization of biomass. In this paper, brewery wastewater sludge (SD) and bituminous coal (BC) were blended for an experimental study which aimed to provide basic experimental data and operational guidance as a reference for practical application in power plants. The co-firing characteristics of sludge and bituminous coal were studied. The results show that the burnout temperature and ignition temperature decrease with an increase in the sludge blending ratio. The Comprehensive Combustion Index (CCI) first rises, then decreases, reaching a maximum at about 15%. Compared with the atmosphere with 79% $N_2$/21% $O_2$, under the 79% $CO_2$/21% $O_2$ atmosphere, ignition is delayed and the burnout temperature is higher. Under an $O_2$/$CO_2$ atmosphere, as the $O_2$ concentration improves, the thermo-gravimetric (TG) curve shifts to the low-temperature region, the burnout temperature drops significantly, and the comprehensive combustion characteristics are improved. With an increment of the heating rate, the curve of TG analysis shifts to the high-temperature region and the CCI increases. This study could provide helpful information on practical blending in coal-fired power plants for energy savings and emission reductions.

**Keywords:** combustion; brewery wastewater sludge; blending; thermo-gravimetric analysis

## 1. Introduction

Over the past few years, with the development of the global economy, energy shortages and environmental pollution have become two important issues facing humankind. The development of efficient and clean combustion of fossil energy, the comprehensive utilization of renewable fuels, and the reduction of environmental pollution have become inevitable requirements for sustainable social development. The brewing industry produces a large amount of distillers' waste of about 100 million tons per year [1] in China. Large amounts of sludge are produced in this process; for example, the wastewater treatment after brewing of Wuliangye Group, Sichuan Province, China, produces sludge with a daily output of about 70–80 tons, which greatly pollutes the environment, causes a waste of resources, and increases the burden on brewing enterprises [2,3]. How to reasonably deal with these sludge has become an urgent problem to be solved.

Blended biomass combustion is one of the effective ways for coal-fired power plants to eliminate solid waste from industrial sources, as it can reduce pollutant emissions, improve fuel flexibility and coal performance, and improve conversion efficiency as well [4,5]. In order to investigate the combustion behavior of blended biomass, abundant research has been carried out regarding ignition, burnout, $NO_X$ emissions, and so on. In addition, the combustion characteristics of coal and biomass fuels, as well as sludge, have been studied extensively by thermo-analytical techniques such as thermo-gravimetric

(TG), derivative thermo-gravimetry (DTG), and differential thermal analysis (DTA) [6,7]. Magdziarz et al. [6] used TG, DTG and DTA to study the combustion of sludge, coal, and two types of biomass, as well as biomass and sludge mixed with coal as their co-firing conditions. Their results showed that the influence of heating rate on the TG curve varies and coal co-combustion with biomass and sludge can obtain economic benefits. Riaza et al. [8] studied coal and biomass fuel's volatile yield and the nitrogen distribution of fuel. They found that most of the fuel nitrogen released by biomass samples is volatile (80–95%), whereas for coal, volatile nitrogen varies considerably with different coal samples. Rui et al. [9] used anthracite and sludge for co-combustion in a fluidized bed burner. The research illustrated that the metal concentration (such as Cu, Ni, Cr, Pb, etc.) of the ash formed by co-combustion is higher and the co-combustion soot shows a lower level of physiological toxicity. Gao et al. [10] used differential scanning calorimetry to study the pyrolysis behavior of dry sludge. The main gases are $CH_4$, CO, $CO_2$, and other volatile organic compounds (VOCs). Except for $CO_2$, the concentration of other gases increases steadily as the pyrolysis temperature increases from 450 to 650 °C. Otero et al. [11] used differential scanning and thermogravimetric analysis to study the difference in characteristics between sludge and coal. Studies have shown that a small sludge blending ratio has a limited effect on coal's heat release and weight loss.

Liu et al. [12] conducted an experimental study on the mixed combustion of coal and dry sludge and concluded that compared with coal, the sludge has two weight loss peaks; after the sludge is blended, the maximum fuel weight loss ratio is advanced and increased. At the same time, the flammability index increases and is between that of the combustion of coal and sludge alone. Hossein et al. [13] used a thermobalance analysis device to reveal the combustion of rice husk and rice straw combined with pre- and post-treatment strategies. The results show that the combustion characteristics of the mixed biomass are improved. Wei et al. [14] used a tubular electric furnace to conduct mixed combustion experiments on municipal sludge and ferro-bituminous coal. The experiments showed that refining the particle size of sludge and coal increases the NO emissions of the blended samples: when the blending ratio of dry sludge and coal is within the range of 10% to 20%, the overall combustion characteristics can be improved and do not increase NO emissions notably.

However, brewery wastewater sludge in Wuliangye Winery at Yibin, Sichuan, China, has a high a calorific value; therefore, it is necessary to experimentally investigate the co-firing of coal and brewery wastewater sludge. In this study, we are mainly concerned about blended coal combustion, brewery wastewater sludge, and bituminous coal. Through analyzing the combustion characteristics of bituminous coal and blended sludge samples, the feasibility of blending sludge in the boiler is determined. This provides the experimental data and a practical reference for the later mixing of sludge in power station boilers and a technical discussion for future $O_2/CO_2$ combustion. The ignition and burnout temperatures of the blended sludge are reduced and the combustion conditions of the appropriate amount of blended sludge can be changed to reduce NO emissions. Due to the improvement in the comprehensive combustion characteristics, coal-fired power stations would have higher combustion efficiency and stability.

## 2. Experiments

### 2.1. Samples

The sample selected for the experiment was the sludge (SD) provided by Wuliangye Winery and the coal type was bituminous coal (BC). In order to study its co-combustion characteristics, during the sample preparation, the sludge and BC were dried and crushed separately, the BC was ground to a particle size of 45–150 μm and the SD was crushed to a particle size of 150–200 μm. Different qualities of coal and sludge were chosen and fully mixed in order to perform co-combustion experiments with different components. They were put into an oven at 105 °C for 24 h for drying pre-treatment [15]. As shown in Table 1, the volatile content of the BC and sludge accounted for 6.78% and 37.72%, respectively. Unlike BC, the main source of the calorific value of sludge is from volatile combustion.

**Table 1.** Proximate and ultimate analysis of the sludge and coal.

| Items | Proximate Analysis (w/%) | | | | Ultimate Analysis (w/%) | | | $Q_{net, ad}/(MJ \times kg^{-1})$ |
|---|---|---|---|---|---|---|---|---|
| | $M_{ad}$ | $A_{ad}$ | $V_{ad}$ | $FC_{ad}$ | $C_{ad}$ | $H_{ad}$ | $N_{ad}$ | |
| BC | 4.78 | 44.66 | 6.78 | 43.78 | 42.26 | 2.12 | 0.52 | 17.12 |
| SD | 2.33 | 59.86 | 37.72 | 0.09 | 17.6 | 2.93 | 2.41 | 6.56 |

$_{ad}$, air dry; M, moisture; A, ash; V, volatile; FC, fixed carbon.

## 2.2. Experimental Methods

A STA449F3 device as a thermo-gravimetric analyzer was adopted for the experiment. The samples were put in an $Al_2O_3$ container with a continuous gas flow, and heated from 50 to 950 °C. The co-firing experiment for sludge and coal was carried out in two atmospheres: one was an $O_2/N_2$ atmosphere and the other was an $O_2/CO_2$ atmosphere. The experimental device is shown in Figure 1.

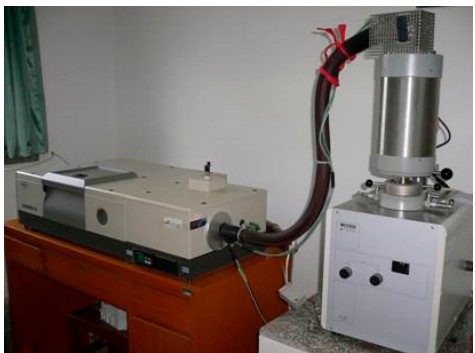

**Figure 1.** The thermo-gravimetric analyzer.

The co-combustion characteristics of diverse blending rates of brewery wastewater sludge are presented and compared. The effects of the physical properties of $N_2$ and $CO_2$ gas on the TG/DTG curve were explored. The effects of different oxygen concentrations on the ignition, burnout, and Comprehensive Combustion Index (CCI) characteristics of the samples under the $O_2/CO_2$ atmosphere were investigated. The CCI indicates the ignition and burnout performance of fuel. The larger the ignition and burnout index, the better the comprehensive combustion behavior. Table 2 shows the experimental cases for this study.

**Table 2.** Experimental cases.

| Experimental Items | Atmosphere | Heating Rate (°C/min) |
|---|---|---|
| SD, BC | | |
| 85% BC + 15% SD | 21% $O_2$/79% $N_2$ | 20 |
| 75% BC + 25% SD | | |
| 50% DG + 50% SD | 21% $O_2$/79% $CO_2$ | 20 |
| | 10% $O_2$/90% $CO_2$ | 20 |
| 15% SD + 85% BC | 21% $O_2$/79% $CO_2$ | 20 |
| | 30% $O_2$/70% $CO_2$ | 20 |
| | 40% $O_2$/60% $CO_2$ | 20 |
| | 21% $O_2$/79% $N_2$ | 10 |
| 15% SD + 85% BC | 21% $O_2$/79% $N_2$ | 20 |
| | 21% $O_2$/79% $N_2$ | 30 |

There are two main indexes for experimental sample combustion: ignition temperature and burnout temperature, which are important parameters to characterize combustion characteristics.

The ignition and burnout temperature can be confirmed by the thermo-gravimetric curve, such as the DTG weight loss rate method, the DTG curve demarcation point method, the tangent method, and so on [16–20]. The ignition temperature refers to the value at which the sample begins to ignite and the ignition temperature can better characterize the difficulty of igniting the fuel. In this study, the tangent method was adopted to determine the ignition temperature of these samples and the first point of a high weight loss rate on the DTG curve was put to use as a vertical line. The TG curve and vertical line have intersection points. The tangent line of the TG curve is made through the intersection point and the temperature corresponding to the intersection point of the tangent line and the smooth baseline after the fuel is dehydrated and dried is determined to be the ignition temperature, $T_i$. The burnout temperature refers to the point at which TG and DTG no longer undergo significant mass changes. $T_b$ is defined by the DTG weight loss rate method: the temperature corresponding to the time when the DTG curve reaches −1 wt%/min at the end of the reaction is defined as the burnout temperature of $T_b$. The CCI reflects the combined combustion of sludge and coal [21] and it contains comprehensive factors of the combustion behavior, including the ignition and burnout temperatures, the maximum DTG reaction rate, and the average DTG reaction rate [22]. A larger CCI indicates that the sample burns more vigorously and rapidly. The following formula [16,21–23] was used:

$$\text{CCI} = \frac{R}{E} \times \frac{d\left(\frac{dW}{dt}\right)_{T=T_i}}{dT} \times \left\{ \frac{(dW/dt)_{max}}{(dW/dt)_{T=T_i}} \right\} \times \frac{(dW/dt)_{mean}}{T_b} = \frac{(dW/dt)_{max} \times (dW/dt)_{mean}}{T_i^2 T_b} \tag{1}$$

where $R$ and $E$ are the coal reactivity index and activation energy (a detailed explanation can be seen in Reference [16]), $(dW/dt)_{max}$ is the maximum reaction rate of DTG, $(dW/dt)_{mean}$ is the average value of DTG between the ignition temperature and burnout temperature in the blended conditions, and $T_i$ and $T_b$ are the ignition and burnout temperatures, respectively.

## 3. Results and Discussion

### 3.1. Combustion Characteristics with Different Sludge Blending Ratios and Atmospheres

Figure 2 shows the DTG and TG curves of sludge and coal in different blending ratios and atmospheres of 21% $O_2$/79% $N_2$ and 21% $O_2$/79% $CO_2$. The figure indicates that the process of co-firing the blending sludge and coal is divided into three stages: dehydration, precipitation of volatiles, and fixed carbon combustion.

As shown in Figure 2, for blended samples, there are two obvious peaks on the DTG curve. With the increase of the sludge blending ratio, the ash content of the sludge is higher than that of coal, so the amount of soot after the blended sample is burned out. The peak value of the volatile matter peak increases significantly and the combustion reaction temperature range becomes larger. The fixed carbon peak-to-peak value decreases and the temperature range of the fixed carbon combustion phase becomes smaller. As the proportion of sludge increases, the weight loss rate of the volatile matter of the blended sample increases and the weight loss rate of the fixed carbon decreases. Both the volatile matter and the fixed carbon precipitate and burn in advance.

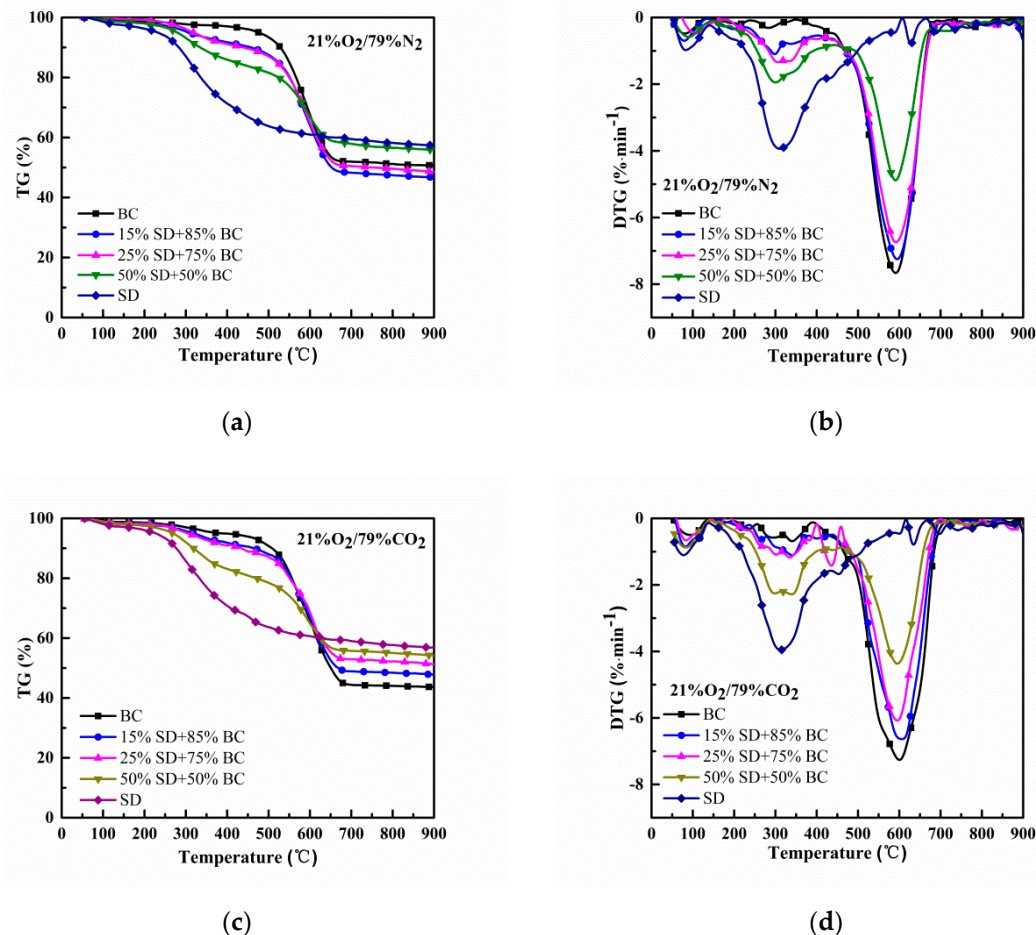

**Figure 2.** Thermo-gravimetric curve of the samples heating in different atmospheres. (**a**) TG curve in 21% $O_2$/79% $N_2$ conditions; (**b**) DTG curve in 21% $O_2$/79% $N_2$ conditions; (**c**) TG curve in 21% $O_2$/79% $CO_2$ conditions; (**d**) DTG curve in 21% $O_2$/79% $CO_2$ conditions.

Compared with the 21% $O_2$/79% $N_2$ atmosphere, the burnout temperature and ignition temperature of the experimental sample under the 21% $O_2$/79% $CO_2$ atmosphere are slightly higher. The burnout temperature addresses the lowest temperature at which the fuel is basically burned out, the main indicator of burnout performance. The lower the burnout temperature, the better the burnout performance. This is because $N_2$ and $CO_2$ have different thermo-physical properties and $CO_2$ has more active chemical properties [24,25]. As the specific heat capacity of $CO_2$ is higher than that of $N_2$, more heat is carried by the airflow, resulting in a lower temperature on the surface of the sludge or coal powder and a decrease in the burning rate. At the same time, the diffusion coefficient of $O_2$ is lower under an $O_2$/$CO_2$ atmosphere and it is more difficult for oxygen to diffuse to the surface and interior of pulverized coal or sludge particles. In addition, when the $CO_2$ concentration is high, the reaction $CO_2 + H \Leftrightarrow CO + OH$ competes with $H + O_2 \Leftrightarrow H + O$ to consume H free radicals, thereby inhibiting the oxidation of CO. In addition, $CO_2$ inhibits $H + O_2 \Leftrightarrow H + O$ by promoting the reaction $H + O_2(+M) \Leftrightarrow HO_2(+M)$, because the latter high three-body collision efficiency reduces the reactivity of the system. Therefore, in the 21% $O_2$/79% $CO_2$ atmosphere, the ignition temperature and burnout temperature of the experimental samples are higher and the CCI is slightly reduced. Reference [2] put forward that increasing the proportion of distillers' grains in coal results in a reduced ignition temperature, burnout temperature, and activation energy in the combustion stage and the Comprehensive Combustion Index (CCI) improves with the same tendency as SD and BC blending.

Figure 3 shows the $T_i$, $T_b$, and CCI of the experimental cases under different atmospheres. The CCI rises first and reaches the maximum when the sludge blending ratio is 15%; that is, $41.13 \times 10^{-8}$

under an air atmosphere and $29.94 \times 10^{-8}$ under an oxygen-rich atmosphere. The CCI value then decreases to $24.52 \times 10^{-8}$ and $21.87 \times 10^{-8}$ when the sludge blending ratio is 50%. This is mainly because a large amount of volatile matter in the sludge in the early stage of combustion pre-ignites and releases heat to promote the combustion of pulverized coal and because the ignition temperature and burnout temperature are reduced, which improves the comprehensive combustion characteristics of the blended fuel. When the sludge blending ratio is large, although the ignition temperature and burnout temperature also decrease, the average combustion rate and maximum combustion rate of the fuel decrease more significantly. At the same time, it can be seen from Table 1 that, compared with the coal containing 44.66% ash, sludge contains 59.86% ash. Therefore, when the sludge blended with fuel increases, the unburned ash of the sludge may be wrapped on the surface of unburned particles, hindering the oxygen diffusing into the particles while restraining the combustion of the inner volatile matter and fixed carbon, making combustion difficult [26].

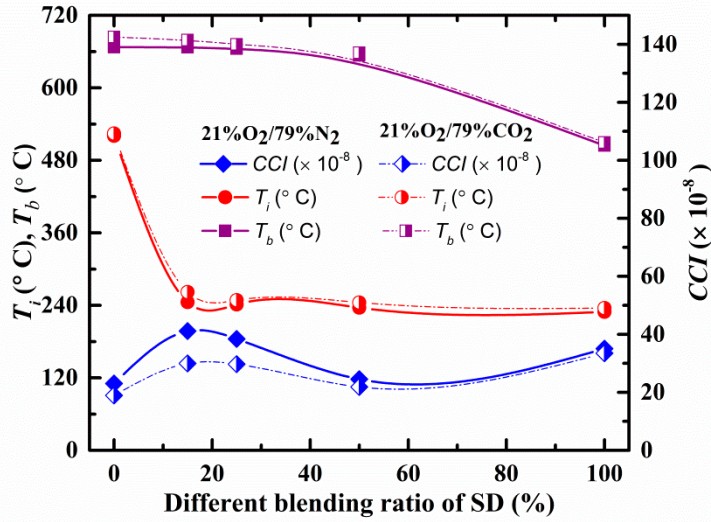

**Figure 3.** Ignition and burnout temperature and CCI at different mixed proportions and combustion conditions.

### 3.2. Effect of the $O_2$ Concentration on Blended Fuels in the $O_2/CO_2$ Atmosphere

Figure 4 shows the thermo-gravimetric curves of different oxygen concentrations under a 21% $O_2$/79% $CO_2$ atmosphere. The figure shows that as the oxygen concentration increases, the thermo-gravimetric curve gradually shifts to the left. This shows that under a high oxygen concentration, the sample burns faster, the reaction temperature range is gradually reduced, and the maximum reaction rate peak gradually increases. Meanwhile, the peak value of the DTG curve increases and the peak corresponding temperature decreases, indicating that the increase of oxygen concentration makes the sample burn at a lower temperature. The combustion peak is reached at a lower point, which is beneficial for combustion. There are two main reasons why the sample burns faster: first, as the oxygen concentration increases, the material comes into contact with oxygen more fully and the combustion is strengthened; second, the increase in oxygen concentration means that the $CO_2$ concentration decreases, which makes the combustion process more efficient. Less heat is taken away and the temperature rises in the furnace, which is more conducive to combustion [3].

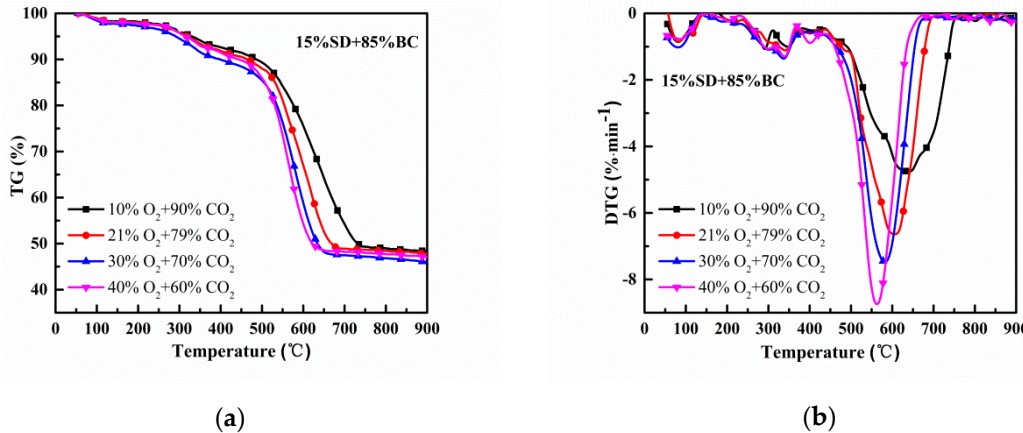

**Figure 4.** Thermo-gravimetric curves of the experimental samples at different O2 concentrations. (**a**) TG curve; (**b**) DTG curve.

Figure 5 shows that as the oxygen concentration increases from 10% to 40%, the ignition temperature $T_i$ of the mixed fuel decreases from 264.13 to 237.30 °C and the burnout temperature $T_b$ decreases from 737.82 °C to 635.85 °C, resulting in comprehensive combustion; the CCI increases as well. However, the decrease in temperature gradually slows as the oxygen concentration increases. This is mainly because when the pulverized coal particles are filled with high-concentration oxygen, the diffusion control zone of the combustion reaction moves to the reaction power control zone and the oxygen concentration is no longer a key factor affecting the combustion characteristics of the blended fuel [3,16]. Compared with Reference [2], with an increase in $O_2$ concentration from 10% to 40%, the ignition temperature changed over a small range but the burnout temperature declined from 745.8 to 641.4 °C, whereas the activation energy and CCI improved. The brewery wastewater sludge has higher combustion temperature characteristics under the $O_2/CO_2$ atmosphere.

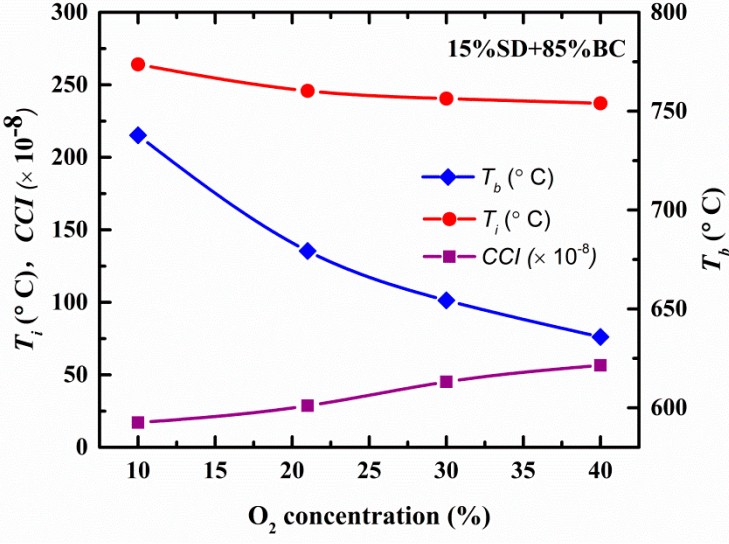

**Figure 5.** Ignition and burnout temperatures and CCI at different $O_2$ concentrations.

### 3.3. Tendency of the Heating Rate of Blended Fuels in the $O_2/N_2$ Atmosphere

Figure 6 shows the thermo-gravimetric curve of the blended samples' combustion at different heating rates. The figure shows that as the heating rate increases, the thermo-gravimetric curve moves to the high temperature zone, whereas at a low heating rate, combustion occurs at a lower temperature. As the heating rate increases, the temperature at which the same conversion rate is achieved is higher. This is because of the "thermal hysteresis" phenomenon [27]; that is, the heating rate is too fast to

cause the external environment's temperature to be high but there is not enough time to transfer heat from the surface of the fuel particles to the inside, resulting in an excessive temperature difference between the inside and outside of the particles, which is not conducive to the pores with formation and diffusion of oxygen. As the heating rate increases, the internal and external heat transfer of the sample is enhanced at higher temperatures, so the combustion becomes more intense (the DTG weight loss rate value increases).

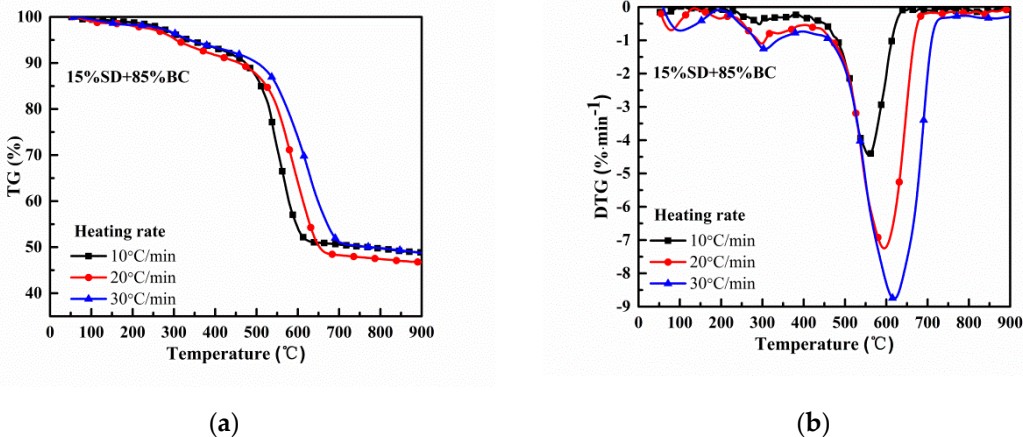

(**a**)　　　　　　　　　　　　　　　　　(**b**)

**Figure 6.** Thermo-gravimetric curves of experimental samples at different heating rates. (**a**) TG curve; (**b**) DTG curve.

Figure 7 shows the ignition temperature, burnout temperature, and CCIs of blended samples with different heating rates. The figure shows that as the heating rate increases (from 10 to 30 °C/min), the ignition temperature increases (from 208.39 to 253.04 °C) but the burnout temperature increases significantly (from 613.11 to 711.06 °C). This is mainly because the "thermal hysteresis" phenomenon occurs when the heating rate is too fast and the blended fuel cannot be completely burnt in the early stage within a short period of time, so the characteristic temperature increases as the heating rate increases. On the whole, as the heating rate rises, the overall combustion characteristics are improved. Aneta (Reference [6]) conducted TG and DTG experiments on coal, wood biomass, oat and sewage sludge blending at different heating rates. They built three steps (regions) of the studied process to reveal the differences in the initial stage of the process from different amounts of moisture and volatile matter and they proposed a criterion for characterizing the ash fouling tendency. This work provided valuable guidance for our next research.

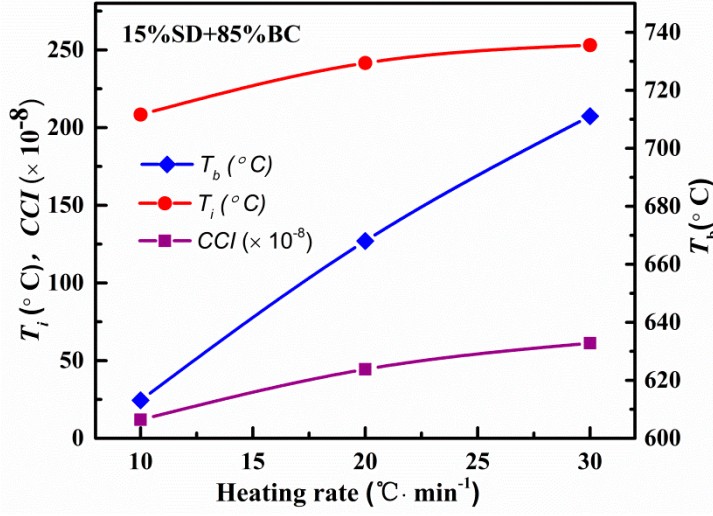

**Figure 7.** Ignition and burnout temperatures and CCI characteristics at different heating rates.

## 4. Conclusions

In this paper, a TG analysis experimental system and a static combustion experimental platform were used to investigate the influence of the sludge blending ratio, atmosphere, $O_2$ concentration, heating rate, and combustion temperature on the combustion characteristics of blended fuel. This provides experimental data and a practical reference for the later mixing of sludge in power station boilers and a technical discussion for future $O_2/CO_2$ combustion. Several conclusions are drawn as follows:

(1) Blending sludge can accelerate the combustion characteristics of single coal. With an increase in the blending ratio, the TG curve shifts to the low-temperature area and the ignition and burnout temperatures of the fuel decrease. Within the 0–50% range of the sludge blending ratio, the burnout temperature only decreases slightly. The ignition temperature drops significantly when the sludge blending ratio is 15%; thereafter, the decline is significantly slower and the CCI reaches the maximum at 15%. It is concluded that the optimal sludge blending ratio is 15% under these experimental conditions.

(2) Under $O_2/N_2$ with 21%/79% and $O_2/CO_2$ with 21%/79% conditions, the change trends of the TG curve of the experimental samples are similar. Compared with the 21% $O_2$/79% $N_2$ conditions, under the 21% $O_2$/79% $CO_2$ atmosphere, ignition is delayed, the burnout temperature is higher, and the CCI is low. In the $O_2/CO_2$ conditions, as the $O_2$ concentration increases, the ignition temperature of the mixed fuel decreases slightly, whereas the burnout temperature decreases significantly. In general, enhancing the oxygen concentration makes the combustion reaction more drastic and improves the CCI.

(3) In the $O_2/N_2$ atmosphere, with an increment of the heating rate, the TG curve moves to the high-temperature zone, the burnout temperature increases significantly, the combustion rate increases, the combustion reaction becomes more intense, and the *CCI* is improved.

In the future, the combustion and operation mode of the boiler can be optimized and adjusted based on the normal operating experience of the boiler. This is of great significance for the reform of China's energy structure, allowing full use of the original thermal power units to be made and, at the same time, realizing the utilization of sludge waste resources, which is a good means of saving environmental costs and creating economic benefits.

**Author Contributions:** Investigation, Y.M.; project administration, Z.L.; resources, Q.C.; writing—original draft, D.X.; writing—review and editing, Z.L. All authors have read and agreed to the published version of the manuscript.

**Funding:** This research is funded by the National Natural Science Foundation of China (No. 51776078) and National key R&D program of China (2019YFC1904003).

**Acknowledgments:** The authors would like to show particular thanks to China Power Fuxi Power Dev. Co. Ltd. for the foundation support and the experimental study.

**Conflicts of Interest:** The authors declare no conflict of interest.

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
