# Peer review of "Experimental Study on Co-Firing of Coal and Brewery Wastewater Sludge"

_applsci, doi:10.3390/app10217589_

Round 1

Reviewer 1 Report

The paper deals with the co-firing of anthracite and brewery sludge. The investigations concerned the co-combustion characteristics in different blending ratios, O2/CO2, and O2/N2 atmospheres, and different oxygen concentrations. The thermogravimetric curve method used in this work is very well known and widely used in the case of co-combustion of coal with various types of raw materials and biomass. While the topic and obtained results are interesting, the introduction and the discussion of the results are prepared rather poorly. Considering these comments and the specific issues listed below, I recommend implementing the corrections below to make the work more meaningful to readers.                                

  1. The introduction is rather poor. As the name suggests, the introduction should introduce the reader to the topic and answer the questions: what is a research problem, what is the reason for the research, what is the purpose of this article? There is no explanation as to why the wastewater sludge brewery has been studied. Why is it worth dealing with the topic of co-firing, what problems are solved by co-firing?
  2. The cited literature sources in the introduction are random; they are quoted without any plan or idea. They only show that anyone has dealt with this topic (e.g. in several places, the mentioned studies concern the NOx emissions during combustion, when the paper does not mention them at all).
  3. Additionally, the authors should clarify the originality and novelty of their research compared to the previously published studies.
  4. In the experiments section there is no explanation of the abbreviations in Table 1: Mad, Aad et al.
  5. Why were the samples ground and the individual fractions separated? What about the rest of the faction types, were they not burned? (line 61-62).
  6. Line 63 – “the volatile content of distiller’s grains” – where is this data in the table? The paper only mentions sludge.
  7. In the Results and discussions section, there are practically no references to similar studies in the literature and no attempts to discuss them.
  8. In my opinion, the information about the practical meaning of testing combustion in O2 / CO2 atmosphere together with the results of other authors would bring significant value to the discussion.
  9. The conclusions are largely a further description of the results, it would be important to limit them to more concise and clear wording.

10. The paper should be rechecked in terms of editing and language (proofreading by a native speaker).

Author Response

Thanks for the reviewer’s valuable comments to greatly improve the paper’s quality.

We have revised the manuscript carefully according to reviewer's helpful advice.

The similarity report entitled for “distillers’ grains/coal” combustion is for coal/grain blending in our research team. We have checked and corrected the pattern in this paper. Some foreign countries’ references have also been regulated to meet the suitable format in the manuscript.

Reviewer 2 Report

In this paper sewage sludge and coal blends were combusted in the different atmosphere using thermogravimetry analysis.

The work is interesting to researchers in this field. However, some major revisions are needed to improve the quality of this work.

  1. The names of cited authors should be rewritten; not Rui Barbosa et al. but Barbosa et al., not Huimin Liuet al. [5] but Liuet et al. [5] etc.
  2. The novelty of paper is not presented clearly what is the gap of knowledge which authors would like to present.
  3. The sentence … there are still many key technical problems that need to be solved urgently, making the technology of blending sludge and coal more standardized and scientific … should be explained in detail. What kind of technical problems authors think about?
  4. In Materials section tested materials are not described enough. There is not mention anything about sludge - type of it (in title is only written that it is from brewery industry) what was the basic parameter before drying etc.
  5. Are authors sure that the tested coal is the anthracite? Parameters of BC didn’t show it Low Heating Value is too low for anthracite and content of ash is too high.
  6. What is it ‘Mad’ in Table 1.
  7. Abbreviation ‘ad’ should be explained under Table 1.
  8. Capture of Fig. 1 is wrong “The thermos-gravimetric ….thermogravimetric should be. Please check hole paper to correct it.
  1. As shown in Table 2.1 …. there is no table nb. 2.1. in the text.
  2. What it is CCI index? What this abbreviation means? Why authors chose it not any others index?
  3. Equation 1 - what is it R, E and other abbreviations.
  4. In Fig. 1 a,b, and Fig 4 a,b lines which belongs to samples should has in the same color like in Fig. 1 b,c and Fig. 2. b,c. It will be more easy to compere decomposition of blends in different atmospheres.
  5. The results of tests are not compared with results of different authors.

Author Response

(The authors gave the same response as above.)

Reviewer 3 Report

The manuscript is about co-firing of coal and brewery wastewater sludge. The scope of this article is consistent with the requirements of the Applied Sciences Journal, but it requires major revision in accordance with the comments below:

  1. I think that introduction is too short. In this part of manuscript the authors should write more about cofiring process.
  2. Please describe exactly, what is the aim of this work.
  3. Line 63: should be table 1, not table 2.1.
  4. All symbols presented in the table 1, should be described.
  5. Table 2: What means the symbol DG? Maybe, it should be SD?
  6. The quality of figure 2 (c, d) is poor.

Author Response

Thanks for the reviewer’s valuable comments to greatly improve the paper’s quality.

We have revised the manuscript carefully according to reviewer's helpful advice.

We have checked and corrected the pattern in this paper. Some foreign countries’ references have also been regulated to meet the suitable format in the manuscript.

Round 2

Reviewer 1 Report

The authors of the article introduced some corrections which made the text more valuable. However, it seems to me that not all of them have been covered yet.

Q5. In the sentence: "the different particle size was selected separately; the anthracite was 45-150 μm, and the sludge was 150–200 μm". "selected separately" suggests that screening and separation of the particles into different fractions has been done. If the division into fractions did not occur, the text in which it is mentioned would have to be re-edited.

Q7: Reply: "We have added corresponding references in results and discussion, and have added some discussions about relative studies."
There are no changes in the Results and discussion section of the submitted version of the article, there are still no references to similar studies by other authors.

Author Response

Q1. In the sentence: "the different particle size was selected separately; the anthracite was 45-150 μm, and the sludge was 150–200 μm". "selected separately" suggests that screening and separation of the particles into different fractions has been done. If the division into fractions did not occur, the text in which it is mentioned would have to be re-edited.

Answer: We would like to thank the reviewer for this helpful comments. The bituminous coal was ground with the particle size of 45-150 μm, and the sludge was crushed with the particle size of 150–200 μm. Different qualities of coal and sludge was chose and fully mixed in order to carry out experiments with different components. According to the reviewer’s helpful suggestion, we have added “the BC was ground into the particle size of 45-150 μm, and the SD was crushed with the particle size of 150–200 μm, then, different qualities of coal and sludge was chose and fully mixed in order to do co-combustion experiments with different components” in the revised manuscript (in section 2.1).

Q2: Reply: "We have added corresponding references in results and discussion, and have added some discussions about relative studies." There are no changes in the Results and discussion section of the submitted version of the article, there are still no references to similar studies by other authors.

Answer: Thanks for your useful advice. Actually, we have modified all the figures in results and discussion section, grammar and some expressions are clarified. At the same time, a native English specialist has corrected the whole manuscript. There are few researches in distillers’ grains wastewater sludge, and this study focuses on the treatment and disposal technologies of resource reuse in brewery wastewater sludge. We try to reveal the experimental research, and plan to carry out in-situ operation in a pulverized coal-fired power plant adjacent to Wuliangye Group. We have added similar studies in municipal sewage sludge by other authors as references from 1 to 8 in the revised manuscript.

Thanks the reviewer for giving some useful directions of the manuscript. We have checked the whole manuscript in detail, and the errors and polish language have been revised.

Reviewer 2 Report

  1. English should be improve by native speaker there is so many mistakes for example ,,,,In china, the….
  2. Authors didn’t extend section Materials, It is still not possible to know details about type of sewage, from which point of process it is coming?
  3. If authors claim that anthracite has calorific value at level 17. MJ/kg please compere your results with parameters of anthracite from literature. From my point it is basic mistake not acceptable for publication to claim that tested coal is called antacid!!
  4. Still index CCI is not explain in the test
  5. Still results of experiments are not compared with results of anther authors

Author Response

  1. English should be improve by native speaker there is so many mistakes for example ,,,,In china, the….

Reply: We appreciate the reviewer’s comments. We have thoroughly examined the whole manuscript, and improved the grammatical mistakes in the revised manuscript. At the same time, the whole manuscript has been corrected a native English specialist.

  1. Authors didn’t extend section Materials, It is still not possible to know details about type of sewage, from which point of process it is coming?

Reply: Thanks for your useful advice. The brewing industry produces a large amount of distillers’ grains waste of about 100 million tons per year in China. Meanwhile, large amounts of brewery wastewater sludge produce in this process, for example, the wastewater treatment after brewing in Wuliangye Group, Sichuan Province, China, produces sludge with a daily output of about 70~80 tons. The components have some differences between distillers’ grains wastewater sludge and municipal sewage sludge. We try to reveal the experimental research, and plan to carry out in-situ operation in a pulverized coal-fired power plant adjacent to Wuliangye Group.

  1. If authors claim that anthracite has calorific value at level 17. MJ/kg please compere your results with parameters of anthracite from literature. From my point it is basic mistake not acceptable for publication to claim that tested coal is called antacid!!

Reply: Thank you for your valuable suggestions, you are very professional, this is a good question. The engineers addressed this kind of coal as anthracite in the foundation supporting company- China Power Fuxi Power Dev. Co. Ltd. for a long time. In fact, it is a bituminous coal according to the coal classification. We have re-examined the previous experimental data, the tested coal is unified as bituminous coal. In addition, we have corrected this mistake in the revised manuscript.

  1. Still index CCI is not explain in the test.

Reply: Thanks. Firstly, in the abstract, “Comprehensive Combustion Index (CCI)” is explained in line 18 on page#1. Secondly, “The CCI reflects the combined combustion of sludge and coal, and it contains comprehensive factors of the combustion behavior including the ignition and burnout temperatures, the maximum DTG reaction rate, and the average DTG reaction rate.” in lines 196-198 on page#4. In addition, we have added “a larger CCI indicates that the sample burns more vigorously and rapidly” in the revised manuscript.

  1. Still results of experiments are not compared with results of anther authors.

Reply: We would like to thank the reviewer for this helpful comments. There are few researches in distillers’ grains wastewater sludge, and relative references by other authors have modified in the revised manuscript. This study focuses on the treatment and disposal technologies of resource reuse in brewery wastewater sludge. We have added similar studies in municipal sewage sludge by other authors as references in the revised manuscript [references 1-8, 15, 16].

Thanks the reviewer for giving some useful directions of the manuscript. We have checked the whole manuscript in detail, and the errors and polish language have been revised.

Reviewer 3 Report

The authors have addressed most of the comments; they have also tried to make changes according to the reviewers’ suggestions. After revisions, the quality of the manuscript has been adequately enhanced. Therefore, the manuscript could be considered for the publication in the Journal.

Author Response

Comments and Suggestions for Authors

The authors have addressed most of the comments; they have also tried to make changes according to the reviewers’ suggestions. After revisions, the quality of the manuscript has been adequately enhanced. Therefore, the manuscript could be considered for the publication in the Journal.

No question.

Round 3

Reviewer 1 Report

In my opinion after revisions the quality of the manuscript has been adequately enhanced, therefore the manuscript could be considered for the publication in the Applied Sciences Journal.

Author Response

In my opinion after revisions the quality of the manuscript has been adequately enhanced, therefore the manuscript could be considered for the publication in the Applied Sciences Journal.

No question.

Reviewer 2 Report

1.Please add to text explanation of  CCI (Comprehensive Combustion Index).

2.Results are still not compared with test results of others authors.

Author Response

Comments and Suggestions for Authors

  1. Please add to text explanation of  CCI (Comprehensive Combustion Index).

Answer: Thanks for your useful advice. We have modified the sentence: “Comprehensive Combustion Index (CCI, indicates the ignition and burnout performance of fuel. The larger the ignition and burnout index, the better the comprehensive combustion behavior)”, and it has been added in the revised manuscript in line 111 on page 3.

  1. Results are still not compared with test results of others authors.

Answer: Thanks for the reviewer’s generous comments. Actually, this study focuses on the treatment and disposal technologies of resource reuse in blending distillers’ grains wastewater sludge with bituminous coal, and there are few researches in brewery wastewater sludge. We try to reveal the experimental research, and plan to carry out in-situ operation in a pulverized coal-fired power plant adjacent to Wuliangye Group, Sichuan Province, China. According to the added similar studies by other authors as references 1-8, 15, 16 in the revised manuscript, we have modified some comparison with distillers’ grains, wood biomass, oat and sludge blending with coal (Ref. 2 and Ref. 6) in result section.

Again, we thank the reviewer for those detailed comments of the manuscript.